# M-LEARNER: A FLEXIBLE AND POWERFUL FRAMEWORK TO STUDY HETEROGENEOUS TREATMENT EFFECT IN MEDIATION MODEL

## ABSTRACT

We propose a novel method, termed the M-learner, for estimating heterogeneous indirect effects and identifying relevant subgroups within a mediation framework. The procedure comprises four key steps. First, we compute individual-level conditional average indirect treatment effect. Second, we construct a distance matrix based on pairwise differences. Third, we apply t-SNE to project this matrix into a low-dimensional Euclidean space, followed by K-means clustering to identify subgroup structures. Finally, we calibrate and refine the clusters using a threshold-based procedure to determine the optimal configuration. This is the first method capable of revealing the complex relationships among X, M, and Y within mediation analysis while effectively controlling the Type I error rate. Experimental results validate the robustness and effectiveness of the proposed framework. Application to the real-world Jobs II dataset highlights the broad adaptability and potential applicability of our method. Code is available at https://anonymous.4open.science/r/M-learner-C4BB.

## 1 INTRODUCTION

Randomized clinical trials are often costly and time-consuming, with significant delays between treatment administration and the observation of final outcomes such as the response variable $Y$. However, mediators can serve as early indicators of treatment efficacy. For example, in colorectal cancer trials, cessation of tumor growth may act as a proxy for positive treatment response. This enables interim evaluation of treatment efficacy and the potential to adapt treatment strategies mid-trial based on changes in the mediator. In practice, treatment effects may be mediated only in a subset of patients. Furthermore, the chosen mediator may not be valid, or treatment effects may be homogeneous (i.e., uniformly effective or ineffective across individuals). Existing approaches—such as the T-learner Künzel et al. (2019)and Causal Random Forest Athey & Imbens (2016)have focused on heterogeneous treatment effect estimation but do not account for mediation mechanisms.

For heterogeneous total treatment effects, Foster et al. (2011) introduced the Virtual Twins to estimate heterogeneous total treatment effects with RCT data. Several methodological approaches within observational studies enable the estimation of flexible and accurate models of heterogeneous total treatment effects. Shalit et al. (2017); Shi et al. (2019); Johansson et al. (2016); Hassanpour & Greiner (2019) use neural networks to estimate heterogeneous total treatment effects. Athey & Imbens (2016); Wager & Athey (2018); Athey & Wager (2019) use random forests to estimate heterogeneous total treatment effects. Künzel et al. (2019) proposed the Meta-learners which consist of S-learner, T-learner, and X-learner. Nie & Wager (2021) proposed R-learner, Zhang et al. (2022) extented the R-learner to continuous treatment scenario. And other methods have been proposed, such as DR-learner and Lp-R-learner (Kennedy, 2023). Dwivedi et al. (2020) proposed model selection solution in causal inference.

When a mediator is present, the primary analytical focus is on evaluating the indirect treatment effect on the outcome via the mediator. These method decompose the average treatment effect into direct treatment effect and indirect effect(Lin et al., 1997; Preacher, 2015; Robins & Greenland, 1992; Petersen et al., 2006; van der Laan & Petersen, 2008; Imai et al., 2010; Tchetgen & Shpitser, 2012; VanderWeele, 2015; Vandenberghe et al., 2017; Dorresteijn et al., 2011; Pearl, 2022; VanderWeele &

Vansteelandt, 2009; Vansteelandt & VanderWeele, 2012; Angrist, 2004; Imbens, 2004). Recently, to address commonly observed intermediated confounders that would be affected by the covariates and then affect both mediators and outcome, multiple methods have been developed to extend the classical metiation analysis(Tchetgen & VanderWeele, 2014; Díaz et al., 2021; 2023; Gilbert et al., 2024). Ge et al. (2023; 2025), Luo et al. (2025) and Wang & Song (2025)proposed the method to use reinforcement learning to deal with the dynamic mediation analysis. Cheng et al. (2022) use deep learning to estimate causal effects in mediation model.

However, these methods do not account for treatment effect heterogeneity in mediation model. Recently, Ting & Linero (2025) proposed a BART-based method for estimating heterogeneous mediation effects, which primarily targets heterogeneous indirect treatment effects. Their approach interprets the estimated indirect treatment effects using decision trees, but it lacks a systematic framework for precise subgroup identification. To bridge these gaps, we propose a flexible and powerful method, termed M-learner, designed to capture heterogeneity in treatment effects mediated by a mediator, we aim to estimate the Conditional Average Indirect Treatment Effect (CAITE) from randomized clinical trial (RCT) data, and to identify subgroups that benefit from treatment by examining heterogeneity in indirect treatment effects. In contrast, our method not only integrates treatment effect heterogeneity into the mediation framework but also provides a systematic strategy for detecting heterogeneous subgroups and regions, with rigorous control of type I error. Crucially, it achieves more accurate subgroup detection by leveraging indirect treatment effects, thereby enabling data-driven decision-making. This enhanced precision allows a clearer characterization of heterogeneity and the contexts in which it arises. For instance, in clinical trials, mediator-derived short-term signals (i.e., indirect treatment effects) can guide treatment allocation to patients most likely to benefit. Our approach facilitates the identification of subgroups exhibiting differential response patterns through mediated pathways, thereby improving the interpretability and clinical relevance of treatment effect estimation. Moreover, our method is more flexible, as it can incorporate a wide range of machine learning algorithms such as random forests, XGBoost, and neural networks. Compared with BART-based approaches, it allows the use of faster algorithms, thereby substantially improving computational efficiency and enhancing scalability to large datasets.

The key contributions of this paper are:

1. We propose the M-learner, a flexible and powerful method for estimating the conditional average indirect treatment effect.

2. We introduce a novel clustering approach for subgroup identification based on indirect treatment effect heterogeneity. By leveraging differences in indirect treatment effects, our method reformulates the unsupervised task of discovering benefiting subgroups into a supervised learning problem.

3. We propose a novel calibration framework to assess the effectiveness of mediators and evaluate the reliability of identified subgroups.

## 2 METHODS

### 2.1 ASSUMPTIONS

Mediation refers to the process by which a treatment (W) affects an outcome (Y) indirectly through an intermediate mediator (M) that lies between the treatment and the outcome (See Figure 1 (a)). In a mediation model, the total treatment effect can be decomposed into two components: the indirect treatment effect (through the mediator, M) and the direct treatment effect (not through M). Here, we focus on the indirect treatment effect. For example, in JOBS II data, the job search self-efficacy is the mediator (M), depressive symptoms is the outcome (Y), receiving job search assistance is the treatment (W). In the JOBS II dataset, we aim to examine whether job search assistance reduces depressive symptoms by enhancing job search self-efficacy.

We have $n$ i.i.d. observations, $\{X_i, Y_i, M_i, W_i\}_{i=1}^n$, for individuals $i$, treatment assignment $W_i = w \in \{0, 1\}$, $X_i = (X_i^{(1)}, \cdots, X_i^{(d)}) \in \mathbb{R}^d$ is a d-dimensional covariate or feature vector, $M_i \in \mathbb{R}$ is the mediator and the outcome $Y_i \in \mathbb{R}$. We denote the potential outcome $M_i(w)$ as the value of the mediator that would have been observed had the individual received treatment $w$. Similarly, $Y_i(w, M_i(w^*))$ is the potential outcome of unit $i$ when $i$ is assigned to treatment $w$, and the mediator

is assigned to treatment $w^*$. We link the potential outcomes to the observed data through the consistency assumption, which states that $M_i = M_i(W_i)$ and $Y_i = Y_i(W_i, M_i(W_i))$. Under these notations, we can define average treatment effects (ATE) and natural direct and indirect treatment effects (DTE, ITE) (Rudolph et al., 2023).

$$
\begin{aligned}
ITE &= \mathbb{E}(Y(1, M(1)) - Y(1, M(0))), DTE = \mathbb{E}(Y(1, M(0)) - Y(0, M(0))), \\
TTE &= \mathbb{E}(Y(1, M(1)) - Y(0, M(0))) = ITE + DTE.
\end{aligned}
$$

In causal mediation analysis, we are particularly interested in estimating the natural indirect treatment effects. The natural indirect treatment effects isolates the effects of the potential mediator in response to different treatment values while keeping the treatment fixed, and can be interpreted as the effects that the treatment has indirectly on the outcome $Y$ through the mediator $M$. We can also define the conditional average indirect treatment effects (CAITE) as $\tau^{ITE}(x) := \mathbb{E}[Y(1, M(1)) - Y(1, M(0))|X = x]$. With the estimation of CAITE, we can use the estimation to study the subgroups of indirect treatment effects. To capture heterogeneity in indirect treatment effects, we partition the support of $\tau^{ITE}(x)$ into a collection of disjoint sets:

$$
\mathbb{R} = \bigcup_{i=1}^{K} \mathcal{U}_i^{ITE}, \mathcal{U}_i^{ITE} \cap \mathcal{U}_j^{ITE}, for\ i \neq j, \tag{1}
$$

where $K$ is unknown. In the absence of heterogeneity, we have $K = 1$. Consequently, subgroups are defined as $\mathcal{G}_i = \{x : \tau(x) \in \mathcal{U}_i^{ITE}\}, \quad i = 1, 2, \dots, K$. We aim to identify heterogeneous treatment subgroups by studying CAITE, so as to inform treatment decisions in clinical trials based on short-term signals reflected through the mediator. So We propose the M-learner, a flexible and powerful framework for estimating CAITE, identify heterogeneous subgroups via mediator. Using this framework, we can identify heterogeneous treatment subgroups and assess the presence of true treatment heterogeneity while controlling the Type I error. Here, we introduce the assumption of the M-learner framework.

For random variables $A$, $B$ and $C$, let $A \perp\!\!\!\perp B|C$ denote that $A$ is conditionally independent of $B$, given $C$, $\mathcal{X} \subseteq \mathbb{R}^d$ denote the sample space of $X$, $\mathcal{M} \subseteq \mathbb{R}$ denotes the mediator space.

**Assumption 1** (Random Treatment Assignment.)**.**

$$
W_i \perp\!\!\!\perp X_i.
$$

*This assumption is fundamental in randomized clinical trials and states that the treatment assignment $W$ is randomly assigned, independent of the covariates $X$.*

**Assumption 2** (Treatment ignorability)**.**

$$
(Y_i(w^*, m), M_i(w)) \perp\!\!\!\perp W_i|X_i = x
$$

*for $w$, $w^* = 0, 1$, and all $x \in \mathcal{X}$. This assumption implies that there are no unmeasured confounders affecting both the treatment and the outcome. Since this is a randomized clinical trial, treatment assignment is independent of baseline covariates. Therefore, Treatment Ignorability naturally holds under Assumption 1 (Random Treatment Assignment).*

**Assumption 3** (Mediator ignorability)**.**

$$
Y_i(w^*, m) \perp\!\!\!\perp M_i(w)|W_i = w, X_i = x
$$

*for all $w$, $w^* = 0, 1$, and all $x \in \mathcal{X}$. In other words, after controlling for $W$ and $X$, the assignment of $M$ is effectively random with respect to the potential outcomes $Y(w, m)$. This assumption is crucial for identifying natural indirect treatment effects in causal mediation analysis.*

**Assumption 4** (Positivity)**.**

$$
f(M_i(w) = m, |W_i = w, X_i = x) > 0
$$

*for all $w, x, m$. $f(M_i(w) = m, |W_i = w, X_i = x)$ denotes the probability density of the potential mediator $M_i(w)$ taking the value $m$ given treatment $W_i = w$ and covariates $X_i = x$. This assumption ensures that, for every combination of treatment and covariates, all mediator values have a non-zero probability. It is necessary for identifying causal mediation effects because it guarantees that we have sufficient support in the data to estimate potential outcomes for all relevant mediator values.*

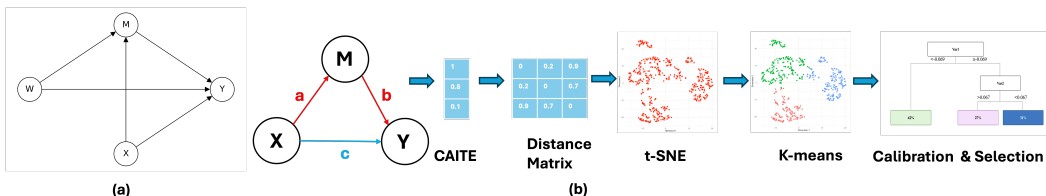

Figure 1: (a)Directed Acyclic Graph between the mediator $M$, treatment $W$, covariates $X$ and response $Y$,(b)the pipeline of the M-learner.

We propose a machine learning–based approach to separately estimate the CAITE, with the goal of identifying subgroups characterized by differences in natural indirect treatment effects. This study addresses several key research questions: How can natural indirect treatment effects be accurately estimated? How can meaningful and interpretable subgroups be identified based on indirect treatment effects heterogeneity? Are the resulting subgroups statistically valid and practically useful? Finally, to what extent is the mediator effective—that is, does a significant portion of the treatment effects operate through the mediator?

## 2.2 M-LEARNER

We propose an algorithm called M-learner[1]shown in Figure 1(b) to address the aforementioned problem. This algorithm consists of four main steps:

1. Estimating conditional average indirect treatment effects,

2. Measuring the difference in treatment effects between pairs of samples (referred to as the " distance"),

3. Projecting the distance matrix into a two-dimensional Euclidean space using t-SNE, and clustering the projected data use K-means,

4. Selecting the optimal clustering result and calibration.

Firstly, we specify the outcome and mediator models:

$$M(w) = \eta_1(x) + w \cdot \kappa_1(x) + \varepsilon, \tag{2}$$
$$Y(w,m) = \eta_2(x) + w \cdot \kappa_2(x) + \kappa_3(x) \cdot M(w) + \zeta, \tag{3}$$

where $\varepsilon, \zeta$ are zero-mean random variables and independent of $X$ and $W$.

**Assumption 5** (Regularity for conditional expectations)**.**

*(i)* $\eta_1(x)$, $\kappa_1(x)$, $\eta_2(x)$, $\kappa_2(x)$, $\kappa_3(x)$ *are measurable in $x$ and finite for all $x \in \mathcal{X}$.*

*(ii)* $\mathbb{E}[|Y||X = x, M = m, W = 1] < \infty$

*(iii)* $\mathbb{E}[\zeta \mid X = x, M = m] = 0$

We denote the treatment mediator function $g_1^M(x) = \mathbb{E}[M|X = x, W = 1]$, control mediator function, $g_0^M(x) = \mathbb{E}[M|X = x, W = 0]$, treatment response function $g_1^Y(x,m) = \mathbb{E}[Y|X = x, M = m, W = 1]$.

**Theorem 2.1** (Consistency)**.** *Under assumptions 1-5, the outcome and mediator model meet equation 2 and equation 3,the CAITE function $\tau^{ITE}(x) = \mathbb{E}[Y(1, M(1)) - Y(1, M(0))|X = x] = g_1^Y(x, g_1^M(x)) - g_1^Y(x, g_0^M(x))$.*

*Proof.*

$$\tau^{ITE}(x) = \mathbb{E}[Y(1, M(1)) - Y(1, M(0))|X = x]$$

---

[1]We refer to the method as the M-learner, as it is specifically designed to operate within the framework of mediation models.

$$
\begin{aligned}
&= \int_{\mathcal{M}} \mathbb{E}[Y|X = x, M = m, W = 1](f(M = m|W = 1, X = x) \\
&\quad - f(M = m|W = 0, X = x))dm \\
&= \int_{\mathcal{M}} g_1^Y(x, m)(f(M = m|W = 1, X = x) - f(M = m|W = 0, X = x))dm \\
&= g_1^Y(x, g_1^M(x)) - g_1^Y(x, g_0^M(x))
\end{aligned}
$$

$\square$

We denote the estimator of $g_1^M$ as $\hat{g}_1^M$, the estimator of $g_0^M$ as $\hat{g}_0^M$, the estimator of $g_1^Y$ as $\hat{g}_1^Y$. The estimation functions can be implemented with a variety of machine learning methods, such as random forests, XGBoost, and neural networks.

So, $\hat{\tau}^{ITE}(x) = \hat{g}_1^Y(x, \hat{g}_1^M(x)) - \hat{g}_1^Y(x, \hat{g}_0^M(x))$. For each unit $i$, we can estimate $\hat{\tau}^{ITE}(x_i)$, denoted as $\hat{\tau}_i$. Now, we have estimated the CAITE, how to get the subgroups of different units? We propose a new method to transform the estimation of CAITE to the clustering.

First, we evaluate the CAITE difference between each pair of units $i$ and $j$, which we refer to as the treatment distance. It is defined by $dis(i, j)$, the distance metric can be defined as Euclidean distance, Manhattan distance, or other formulations. In the following analysis, we consistently adopt Euclidean distance to compute the distance between units $i$ and $j$ which means $dis(i, j) = (\hat{\tau}_i - \hat{\tau}_j)^2$. Consequently, an $n \times n$ distance matrix is obtained. In fact, the distance matrix preserve the local similarity between different units. This matrix is then projected into a two-dimensional Euclidean space using t-SNE(Van der Maaten & Hinton, 2008), where each unit $i$ is assigned a coordinate, The detailed steps of t-SNE can be found in the Appendix A.1. The purpose of this step is to project the data into a two-dimensional space while preserving local similarities. Alternative algorithms such as UMAP also exist, but t-SNE is generally regarded as one of the most effective methods for maintaining local structure during projection, a point we further elaborate on in the Appendix A.4.3 and A.4.4. The projected space from t-SNE often results in approximately spherical clusters (Kobak & Berens, 2019), a setting where K-means is known to work well. So, K-means clustering is performed on the projected points in the Euclidean space. The range of cluster numbers (from 2 to $k$) for K-means clustering must be specified in advance, based on the sample size and the number of covariates; an excessively large $k$ can result in substantial overfitting, based on our experience, we recommend $k = \lfloor \sqrt{d} \rfloor + 2$, where $\lfloor \rfloor$ represents rounding down to the nearest integer. A comprehensive discussion on the selection of the number of clusters $k$ is provided in the Appendix A.4.5. Then, a decision tree is employed to model the clustering results obtained for each predefined number of clusters, with the objective of mapping the unknown categories to interpretable categorical information. For each leaf of the decision tree, we refer to it as a subtype. We use $p_{leaf}$ to select the final, unique subtype grouping from different decision tree results. The definition of $p_{leaf}$ is as follows,

$$
\begin{aligned}
M &= \beta_1 \text{leaf} + \beta_2 W, & (4) \\
M &= \beta_3 \text{leaf} + \beta_4 W + \beta_5 \text{leaf} * W, & (5)
\end{aligned}
$$

the likelihood functions of 4 and 5 are $L_0$ and $L_1$, respectively. Then $2(\log L_1 - \log L_0)$ follows a chi-squared distribution,where the degrees of freedom correspond to the number of decision tree leaves minus 1. Based on this, we calculate $p_{leaf}$. The decision tree result with the minimal $p_{leaf}$ is chosen as the final subtype classification.

## 3 SIMULATION

In real-world datasets, ground truth causal treatment effects are rarely directly observable. Consequently, empirical evaluation of causal inference methods often relies on synthetic data. For such evaluations to yield meaningful conclusions, the synthetic data must closely reflect real-world characteristics. In all experiments, we employed Random Forests (RF) and XGBoost (XGB) as base learners(Breiman, 2001; Chen & Guestrin, 2016). Unless otherwise stated, all experiments in this section are conducted with a sample size of 1000 and 10 covariates.

To identify subgroups that benefit from the treatment effects through the mediator, we design five scenarios that reflect varying real-world heterogeneity structures. Specifically, the scenarios represent

(i) existing heterogeneity, all treatment effect via mediator (All); (ii) existing heterogeneity,part treatment effect via mediator (Part); (iii) no heterogeneity, no treatment effect via mediator (Null1); (iv) no heterogeneity, no treatment effect via mediator (Null2); (v) no heterogeneity, all units benefit from the treatment and all treatment effect via mediator (Global); We calibrate the threshold based on Scenario Null2 by controlling the Type I error at $10\%$, and then apply this threshold to assess the validity of subtype identification across other scenarios, further experimental details are available in the Appendix A.2.

Table 1 presents the results from 100 replicated simulations of the mediation model, using two different base learners, and reports the frequency with which covariates $X^{(1)}$ and $X^{(2)}$ are included in the identified final subtypes. Figure 2 presents the rejection rates of our method across different scenarios after calibration. The results demonstrate that our approach effectively controls the type I error rate, thereby reducing the risk of erroneous decisions and substantially lowering the likelihood of misclassification in intervention choices. Table 2 presents a comparison between the subtypes identified by the M-learner and the true heterogeneous regions, based on sample size and mediation proportion, across two heterogeneous scenarios. As shown in Table 2, the subtype regions identified by the M-learner are smaller than the true heterogeneous regions; however, the estimated mediation proportions closely approximate those of the true regions. Figure 4 shows the heterogeneous region thresholds identified by M-learner. The results shows that our method is powerful to support the causal decision making. The empirical cumulative distribution functions (ECDF) of the p-values under different scenarios are illustrated in Figure 3, the figures elucidates why the $p_{leaf}$ is instrumental in discerning non-heterogeneous scenarios.

Table 1: The table summarizes the correct covariates in profiles in each scenario. $X^{(1)}/X^{(2)}$:Final profile contain $X^{(1)}/X^{(2)}$, $X^{(1)}\&X^{(2)}$: final profile contain both $X^{(1)}$ and $X^{(2)}$, $X^{(1)}$ and $X^{(2)}$ are the two variables associated with treatment effect heterogeneity. Each value denotes the count of occurrences across 100 simulations.

| Covariates | Random Forest | | | XGBoost | | |
|:---:|:---:|:---:|:---:|:---:|:---:|:---:|
| | $X^{(1)}$ | $X^{(2)}$ | $X^{(1)}\&X^{(2)}$ | $X^{(1)}$ | $X^{(2)}$ | $X^{(1)}\&X^{(2)}$ |
| All | 99 | 99 | 98 | 100 | 100 | 100 |
| Part | 100 | 99 | 99 | 100 | 100 | 100 |
| Null1 | 9 | 6 | 4 | 8 | 4 | 2 |
| Null2 | 4 | 4 | 1 | 2 | 1 | 0 |
| Global | 2 | 2 | 1 | 3 | 5 | 1 |

Table 2: Comparison of true and M-learner–estimated mediation proportions and sample sizes within heterogeneous treatment effect regions across scenarios. Standard deviations are shown in parentheses. Med prop denotes mediation proportion:=indirect treatment effect/ total treatment effect. The mediation proportion is calculated by R package "mediation".

| Scenario | Ground Truth | | Random Forest | | XGBoost | |
|:---:|:---:|:---:|:---:|:---:|:---:|:---:|
| | N | Med Prop | N | Med Prop | N | Med Prop |
| All | 251(13) | 1.37(0.09) | 214(46) | 1.33(0.10) | 229(50) | 1.33(0.11) |
| Part | 251(13) | 0.76(0.03) | 219(47) | 0.78(0.06) | 221(39) | 0.78(0.07) |

In different scenarios, both learners are capable of effectively identifying subgroups reflecting treatment effects transmitted through the mediator. However, from Figure 4, XGBoost consistently outperforms Random Forest. Specifically, the M-learner framework with XGBoost demonstrates strong capacity to detect indirect treatment effects and to recover meaningful subgroups accordingly. It also robustly rejects cases where the indirect effect is zero, indicating an ineffective mediator. Null1 and Null2 both assume a null indirect effect, but differ in whether the mediator influences the outcome. In Null1, the treatment does not affect the mediator, while in Null2, the mediator has no association with the outcome, implying it is not a true mediator. Our proposed method successfully identifies both types of scenarios using either base learner. Overall, the experimental results strongly support our hypotheses: across varying degrees of complexity, the proposed approach reliably determines the effectiveness of the mediator $M$, estimates the mediation effect, and identifies the corresponding subgroups.

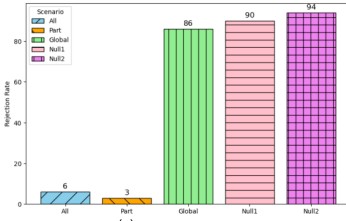 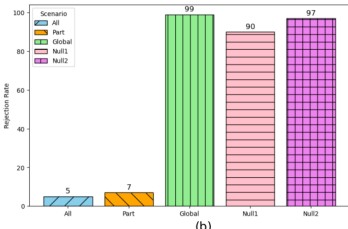

Figure 2: This table reports, after calibration, the number of rejections across 100 replications for each scenario, corresponding to the cases where no heterogeneous subgroups are identified. (a) RF learner, (b) XGB learner.

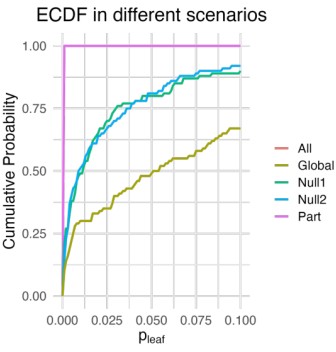 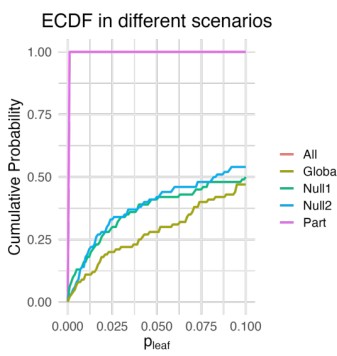

Figure 3: Empirical cumulative distribution functions (ECDF) of $p_{leaf}$ under five different scenarios. (a) RF learner, (b) XGB learner.

We compared our algorithm with BART method(Ting & Linero, 2025), the detailed results see Appendix A.3. In addition, we evaluated the sensitivity of the proposed method with respect to various factors, including different noise levels, sample size, projection dimensions, projection techniques, and the number of clusters. A detailed discussion and corresponding results are provided in the Appendix A.4.

## 4 REAL DATA APPLICATION

In this section, we apply the proposed method to analyze the JOBS II real dataset, collected from a randomized of a job training intervention on unemployed workers. The dataset can be downloaded from R package "mediation"(Tingley et al., 2014). The JOBS II study is a well-known randomized controlled trial conducted in the United States, designed to evaluate the effectiveness of a job search intervention program for unemployed individuals. The study enrolled 899 participants, who were randomly assigned either to a job training intervention group or to a control group. The dataset includes a rich set of covariates, such as demographic information and psychological measures (e.g., self-efficacy, depression). In follow-up interviews, the outcome—a continuous measure of depressive symptoms was assessed. The mediator $M$, is a continuous measure representing job search self-efficacy(Price et al., 1992; Vinokur et al., 2000; Vuori & Silvonen, 2005; Cheng et al., 2022).

We use age, sex, education, prior occupational status, and the level of economic hardship experienced by participants as covariates. For educational background, we classify participants into three categories: individuals who did not complete high school or whose highest degree is a high school diploma are grouped together; individuals who attended some college but did not obtain a bachelor's degree form the second group; and those who earned a bachelor's degree or higher are classified into the third group. Other covariates are kept in their original form.

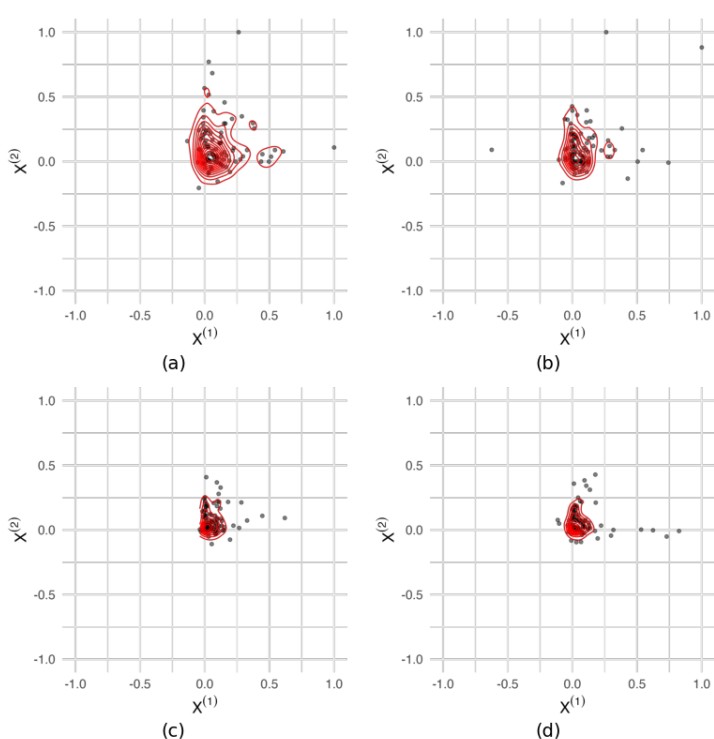

Figure 4: Threshold location distribution plots. dot in the figure represents the threshold of heterogeneous region for 100 replications, red line represents the density line. If a variable was not selected or the threshold exceeded 1, we assigned a value of 1. (a) RF leaner, All scenario, (b) RF learner, Part scenario, (c) XGB learner, All scenario, (d) XGB learner, Part scenario.

We first conducted a preliminary analysis of the JOBS II data using the mediation package(Imai et al., 2010). Table 3 shows the detailed analysis. Job search self-efficacy has been widely employed as a mediator in the literature(Vinokur et al., 1995; 2000). However, it exhibits relatively large p-value.

Applying the proposed M-learner to the JOBS II dataset, as illustrated by the decision tree in Figure 5, identified two distinct subgroups. As presented in Table 3, subtype 1 demonstrated significant TTE and ITE, accompanied by a substantial proportion of mediation and a reduced p-value. Conversely, subtype 2 exhibited smaller and statistically non-significant TTE and ITE.

Table 3: Summary of the total treatment effect, indirect treatment effect, mediation proportion, and corresponding p-values, calculated using methods from the "mediation" package Imai et al. (2010)and the proposed approach. Med prop denotes mediation proportion:=indirect treatment effect/ total treatment effect. TTE denotes total treatment effect. ITE denotes indirect treatment effect. The value in parentheses is the p-value. N denotes sample size.

| Method | Subtype | N | TTE | ITE | Med Prop |
|---|---|---|---|---|---|
| "Mediation" | NA | 899 | $-0.063(0.29)$ | $-0.015(0.21)$ | $24.0\%(0.29)$ |
| M-learner | Subtype1 | 534 | $-0.15(0.016)$ | $-0.030(0.010)$ | $20.0\%(0.052)$ |
| | Subtype2 | 365 | $0.065(0.30)$ | $0.005(0.75)$ | $8.0\%(0.74)$ |

These results suggest that the mediator exhibits heterogeneity across different subtypes and serves as an effective mediator only in certain groups. Using this approach, in future government-initiated randomized clinical trials, it would be possible to monitor changes in the mediator within specific subtypes to quickly assess whether a new intervention is effective for those groups. Trials could be stopped early for ineffective interventions in particular groups, allowing better, more targeted treatments to be administered. This strategy can significantly reduce government costs while enabling

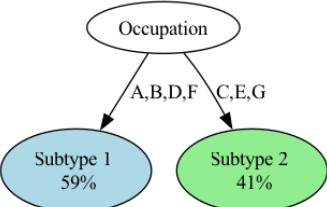

Figure 5: Subtype results identified by the M-learner on the Jobs II dataset. In the decision tree, each leaf represents a subtype. econ_hard refers to the level of economic hardship. In occupation, A denotes "clericalkindred",B denotes "laborersservice works",C denotes "operativeskindred works",D denotes"sales workers",E denotes "craftsmenforemenkindred",F denotes "manegerial", G denotes "professionals".In education, A denotes individuals who did not complete high school or whose highest degree is a high school diploma, B denotes individuals who attended some college but did not obtain a bachelor's degree, C denotes those who earned a bachelor's degree or higher.

timely adjustments to intervention strategies, thereby minimizing the negative impact of ineffective treatments on participants. Our proposed M-learner method effectively uncovers heterogeneous subgroups in the indirect treatment effect. It not only detects heterogeneity but also validates the identified subgroups while controlling the type I error rate, which is of substantial practical importance.

## 5 DISCUSSION

This paper introduces a flexible and powerful method for estimating and uncovering the complex relationships among $X$, $W$, $M$, and $Y$ in causal mediation analysis. In addition to estimating mediated heterogeneity, our approach enables data-driven subgroup identification based on distinct mediator patterns. This represents a significant methodological advancement, as it allows researchers to evaluate the informativeness of mediators and to detect heterogeneous treatment effects that are revealed through them. The proposed framework is highly flexible and can accommodate a variety of base learners, including random forests, XGBoost, and neural networks, depending on the data characteristics and specific application context. The method estimates the CAITE and identifies heterogeneity and subgroup structures that arise through the mediator. By capturing treatment effect heterogeneity from mediated perspectives, the framework offers a comprehensive understanding of the complex interplay among treatment, covariates, mediators, and outcomes.

Although our method is developed under the assumption of an RCT, it can be extended to observational studies by incorporating appropriate adjustments using propensity scores.

Unlike Ting & Linero (2025), which focuses primarily on estimating CAITE, our method not only estimates CAITE but also enables accurate subgroup identification to support decision-making. In addition, our approach is more flexible, allowing for the use of different learners to accommodate various scenarios. Furthermore, our approach effectively controls the type I error rate, allowing for the reliable exclusion of NULL and Global scenarios—an aspect critical for decision-making. In addition, it offers substantially higher computational efficiency. For instance, in our simulation experiments, estimating CAITE with XGBoost for a single replication takes less than one second, compared to 10 minutes required by the BART (single CPU core of an M3 Max).

Importantly, our method permits the inference of individual treatment responsiveness without requiring observation of the final outcome variable $Y$. This finding holds substantial implications across disciplines such as economics, psychology, medicine and sociology. In practical applications, such as in technology companies, mediator behavior can guide personalized interventions. In healthcare, particularly in pharmaceutical settings, the method can assist clinicians in adapting treatment strategies, thereby contributing to the development of precision medicine.

A potential direction for future research is to extend our method to survival analysis settings, which could enable the pharmaceutical industry to predefine subgroups based on mediating variables, ultimately enhancing the success rate of drug development.

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

# A APPENDIX

## A.1 DETAILED OF T-SNE PROCESS

Here, we introduce the process of t-SNE projection in our method. To simplify the notation, we denote $dis(i, j)$ by $d_{ij}$.

In t-SNE projection process, the first step is to calculate the similarity probability:

$$p_{j|i} = \frac{\exp\left(-d_{ij}^2/2\sigma_i^2\right)}{\sum_{k\neq i}\exp(-d_{ik}^2/2\sigma_i^2)} \tag{6}$$

where $p_{j|i}$ is the similarity probability, $\sigma_i$ is determined by preplexity, which is chosen by user, the detailed introduction can be found in (Maaten & Hinton, 2008; Cai & Ma, 2022). Then symmetric similarity probability

$$p_{ij} = \frac{p_{j|i} + p_{i|j}}{2n},$$

Then, defined the similarity in low dimension (Euclidean space),

$$q_{ij} = \frac{(1 + \|o_i - o_j\|^2)^{-1}}{\sum_{k\neq i}\exp(1 + \|o_k - o_l\|^2)^{-1}},$$

where $o_i$ represents the coordinates of the $i$-th individual in the Euclidean embedding.

Finally, t-SNE optimizes the Kullback–Leibler divergence

$$\min \sum_{i \neq j} p_{ij} \log \frac{p_{ij}}{q_{ij}}$$

to determine the coordinates of different units in the Euclidean space.

### A.2 SETTING IN MEDIATOR MODEL

Across all experimental designs in mediator model, we fixed the sample size at $1,000$, with $10$ covariates generated for each unit. Subjects were randomly assigned to treatment and control groups in a $1:1$ ratio. All functions were defined for units under treatment ($w = 1$) and control ($w = 0$),

$$
\begin{aligned}
M_i(w) &= \eta_1(X_i) + w \cdot \kappa_1(X_i) + \varepsilon_i, & (7) \\
Y_i(w) &= \eta_2(X_i) + w \cdot \kappa_2(X_i) + \kappa_3(X_i) \cdot M_i(w) + \zeta_i, & (8)
\end{aligned}
$$

where $\varepsilon_i \sim \mathcal{N}(0, 0.01)$, $\zeta_i \sim \mathcal{N}(0, 0.01)$, and the $X_i$ are independent of $\varepsilon_i$, $\zeta_i$ and one another, and $X_i^{(j)} \sim \mathcal{N}(0, 1)$, for $j = 1, \cdots, d$.

The five scenarios designs follow:

1. In (7), existing heterogeneity,all treatment effects via mediator (All):$\eta_1(X) = \frac{1}{2}(X^{(1)} + X^{(2)}) + X^{(3)} + X^{(4)}$,$\kappa_1(X) = \sum_{i=1}^{2} \mathbb{I}(X^{(i)} > 0) \cdot X^{(i)}$, $\eta_2(X) = \frac{1}{2}(X^{(3)} + X^{(4)}) + 1$,$\kappa_2(X) = 0$, $\kappa_3(X) = 1$.

2. In (7), existing heterogeneity, part of the treatment effect via mediator (Part): $\eta_1(X) = \frac{1}{2}(X^{(1)} + X^{(2)}) + X^{(3)} + X^{(4)}$,$\kappa_1(X) = \sum_{i=1}^{2} \mathbb{I}(X^{(i)} > 0) \cdot X^{(i)}$, $\eta_2(X) = \frac{1}{2}(X^{(3)} + X^{(4)}) + 1$,$\kappa_2(X) = \sum_{i=1}^{2} \mathbb{I}(X^{(i)} > 0) \cdot X^{(i)}$, $\kappa_3(X) = 1$.

3. In (7), no heterogeneity, $0\%$ treatment effects via mediator(NULL 1): $\eta_1(X) = \frac{1}{2}(X^{(1)} + X^{(2)}) + X^{(3)} + X^{(4)}$,$\kappa_1(X) = 0$, $\eta_2(X) = \frac{1}{2}(X^{(3)} + X^{(4)}) + 1$,$\kappa_2(X) = \sum_{i=1}^{2} \mathbb{I}(X^{(i)} > 0) \cdot X^{(i)}$, $\kappa_3(X) = 1$.

4. In (7), no heterogeneity, $M$ is not mediator, all treatment effects are directly transmitted to $Y$(NULL 2):$\eta_1(X) = \frac{1}{2}(X^{(1)} + X^{(2)}) + X^{(3)} + X^{(4)}$,$\kappa_1(X) = 0$, $\eta_2(X) = \frac{1}{2}(X^{(3)} + X^{(4)}) + 1$,$\kappa_2(X) = \sum_{i=1}^{2} \mathbb{I}(X^{(i)} > 0) \cdot X^{(i)}$, $\kappa_3(X) = 0$.

5. In (7), no heterogeneity, all units benefit from the treatment and all treatment effects via mediator (Global):$\eta_1(X) = \frac{1}{2}(X^{(1)} + X^{(2)}) + X^{(3)} + X^{(4)}$,$\kappa_1(X) = 1$, $\eta_2(X) = \frac{1}{2}(X^{(3)} + X^{(4)}) + 1$,$\kappa_2(X) = 0$, $\kappa_3(X) = 1$.

For the RF model, the number of trees is set to 2000, with all other parameters kept at their default values. For XGBoost, the number of boosting rounds is set to 100, while all remaining parameters are left at their default settings. In all simulation experiments, the range of cluster numbers is predefined as 2 to 5.

In this paper, all experiments were conducted on a MacBook Pro equipped with an M3 Max CPU and 36GB of RAM. The software environment includes R version 4.3.3, with the randomForestSRC package version 3.2.3, xgboost version 1.7.9.1 and rpart version 4.1.23.

### A.3 COMPARATIVE ANALYSIS WITH EXISTING METHODS

Recently, Ting & Linero (2025) developed a method for estimating the conditional average indirect treatment effect, which is based on BART, and use decision tree to explain the final results. Throughout this section, we denote this method as BART. In this experiment, the M-learner is implemented with XGBoost serving as the base learner. All experimental parameters are set same as specified in the original paper, without modification for BART method.

Table 4 shows that, based on the RMSE criterion, BART outperforms the M-learner in estimating CAITE under the heterogeneous scenarios (All and Part scenario). In contrast, the two methods perform comparably in the Null1 and Null2 scenarios, while in the Global scenario, the M-learner achieves better performance than BART. Ting & Linero (2025) use CART summary of the posterior mean of CAITE to get subtypes. In Ting & Linero (2025), different leaf nodes are treated as different subtypes. To compare subgroup results across the two methods, we fit a linear model of the outcome $Y$ on the treatment indicator (TRT) within each leaf of the decision tree. The leaf with the largest treatment effect is regarded as the heterogeneous subgroup region. We then extract the corresponding split thresholds that define this region and visualize them in Figure 6 and 7. These results indicate that, compared with BART, the M-learner can more effectively identify heterogeneous subgroups. In contrast, BART tends to overfit the subgroup results, whereas our method is better suited for decision-making and more reliably uncovers meaningful subgroups.

Table 4: An RMSE-based comparison of M-learner and BART approaches for estimating CAITE.

| Scenario | BART | M-learner |
|----------|------|-----------|
| All      | 0.16 | 0.29      |
| Part     | 0.44 | 0.63      |
| Null1    | 0.90 | 0.83      |
| Null2    | 0.91 | 0.89      |
| Global   | 0.86 | 0.51      |

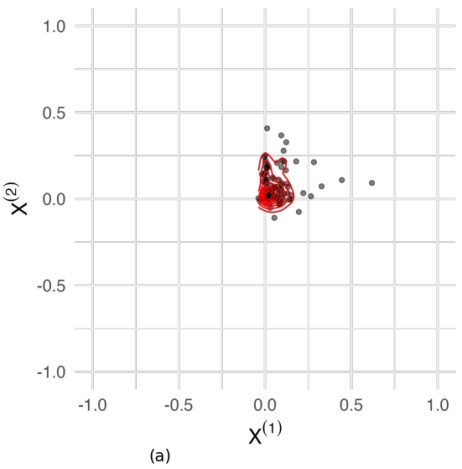 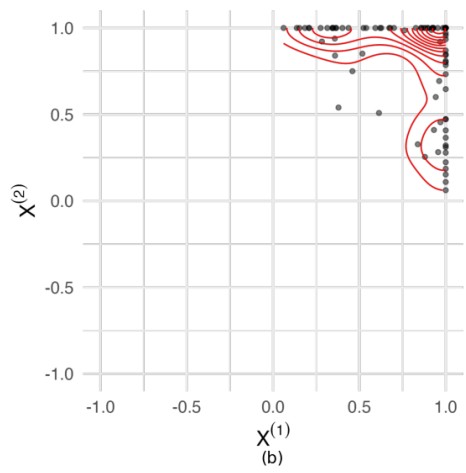

Figure 6: Threshold location distribution plots. dot in the figure represents the threshold of heterogeneous region for 100 replications, red line represents the density line. If a variable was not selected or the threshold exceeded 1, we assigned a value of 1. (a)M-learner, All scenario, (b) BART, All scenario.

To further compare the ability of the two methods to capture heterogeneous subgroups, we conducted an additional experiment. For both methods, we applied the clustering procedure proposed in this paper: specifically, we computed the treatment distance matrix, applied t-SNE for dimensionality reduction, performed K-means clustering, and then identified subgroups based on $p_{leaf}$ (without calibration). Thus, the same clustering procedure was employed in both cases, with the only difference being the approach used to estimate the CAITE.

The results in Figure 8 and 9 indicate that BART exhibits considerably lower accuracy in threshold selection compared to the M-learner. Combined with the RMSE-based evaluation, these findings suggest that the estimates from BART show signs of overfitting. By contrast, the M-learner more effectively distinguishes heterogeneous from non-heterogeneous groups. While RMSE is a global metric that reflects the model's overall average fitting accuracy, it is less directly aligned with decision-making tasks, such as subgroup identification, where the key objective is to clearly separate distinct

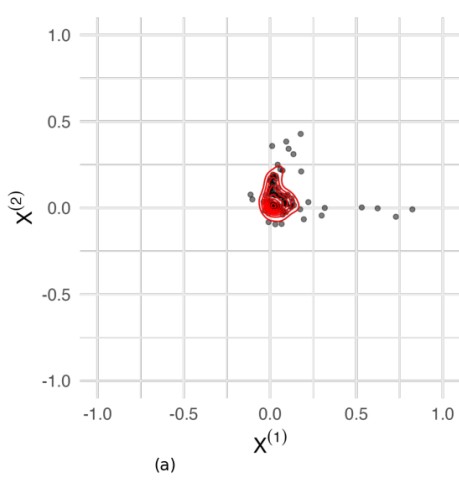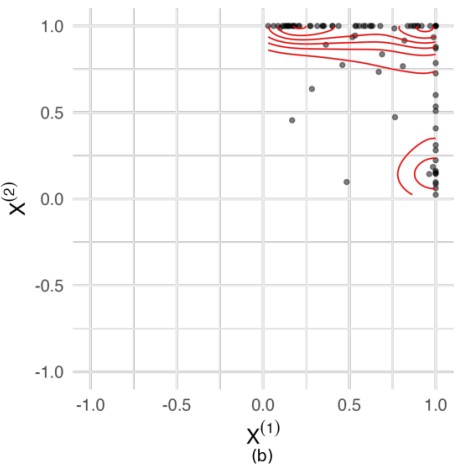

Figure 7: Threshold location distribution plots. dot in the figure represents the threshold of heterogeneous region for 100 replications, red line represents the density line. If a variable was not selected or the threshold exceeded 1, we assigned a value of 1. (a)M-learner, Part scenario, (b) BART, Part scenario.

populations. This experiment provides strong evidence of the superior ability of our method to discriminate between heterogeneous indirect treatment effects.

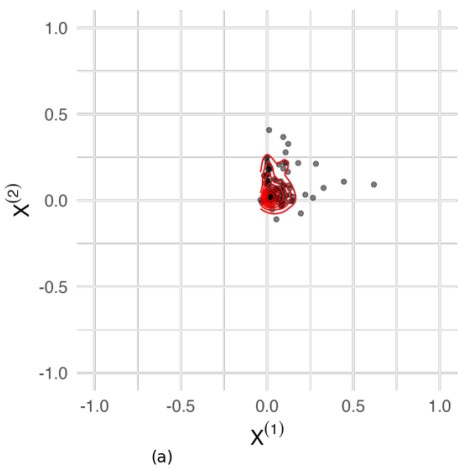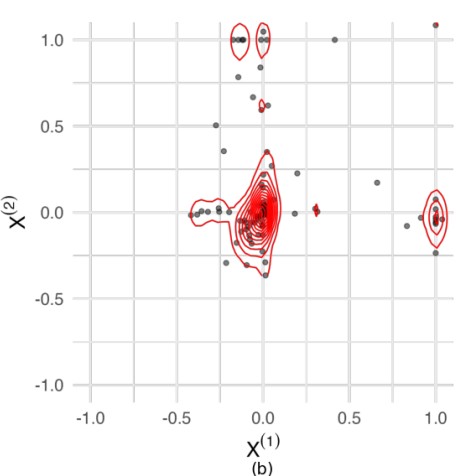

Figure 8: Threshold location distribution plots. dot in the figure represents the threshold of heterogeneous region for 100 replications, red line represents the density line. If a variable was not selected or the threshold exceeded 1, we assigned a value of 1. (a)M-learner, All scenario, (b) BART, All scenario.

## A.4 SENSITIVITY ANALYSIS

### A.4.1 SAMPLE SIZE

To assess the performance of the M-learner under varying sample sizes, we additionally compare the results to those obtained with a sample size of $500$, while holding all other conditions consistent, and in this experiment, the base learner is XGBoost. Figure 10 and Table 5 show that M-learner is still powerful in smaller dataset, the performance only exhibits a slight decline, suggesting that our method remains effective for supporting decision-making in practice, even when applied to relatively small sample sizes.

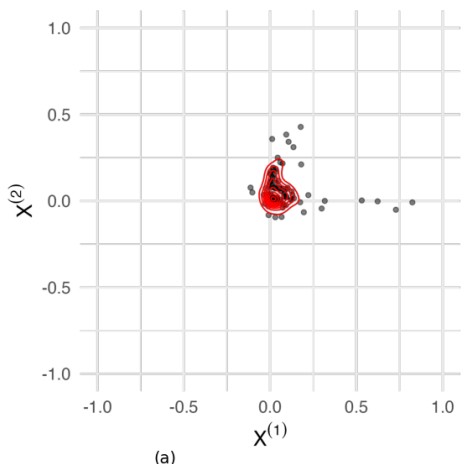
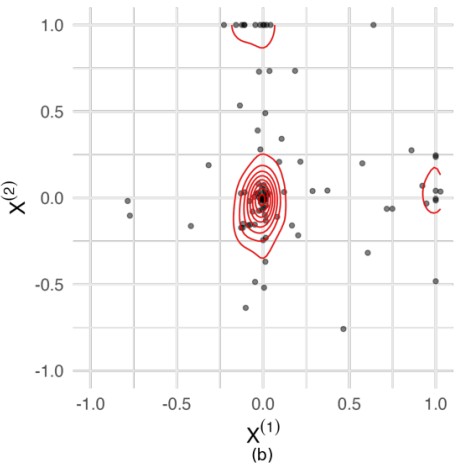

Figure 9: Threshold location distribution plots. dot in the figure represents the threshold of heterogeneous region for 100 replications, red line represents the density line. If a variable was not selected or the threshold exceeded 1, we assigned a value of 1. (a)M-learner, Part scenario, (b) BART, Part scenario.

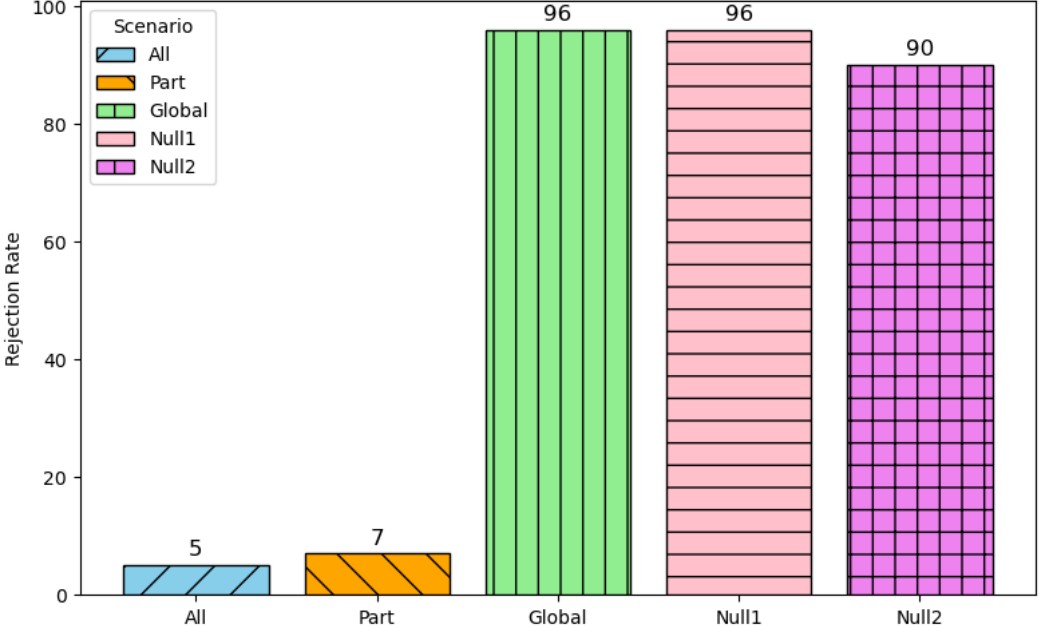

Figure 10: This table reports, after calibration, the number of rejections across 100 replications for each scenario when sample size is 500, corresponding to the cases where no heterogeneous subgroups are identified.

Table 5: The table summarizes the correct covariates in profiles in each scenario when sample size is 500. $X^{(1)}/X^{(2)}$:Final profile contain $X^{(1)}/X^{(2)}$, $X^{(1)}\&X^{(2)}$: final profile contain both $X^{(1)}$ and $X^{(2)}$, $X^{(1)}$, $X^{(2)}$ are the two variables associated with treatment effect heterogeneity. Each value denotes the count of occurrences across 100 simulations.

| Covariates | $X^{(1)}$ | $X^{(2)}$ | $X^{(1)}\&X^{(2)}$ |
|---|---|---|---|
| All | 92 | 94 | 91 |
| Part | 92 | 93 | 92 |
| Null1 | 0 | 1 | 0 |
| Null2 | 2 | 2 | 1 |
| Global | 0 | 2 | 0 |

### A.4.2 Noise

To evaluate the robustness of the proposed method under varying levels of noise, we introduced additive Gaussian noise with different variances. The noise levels were categorized as follows (both $\varepsilon_i$ and $\zeta_i$ in equation 7),:

- low noise (original setting): $N(0, 0.01)$;
- moderate noise: $N(0, 0.1)$;
- high noise: $N(0, 1)$.

All other experimental settings were kept identical to All scenario in A.2, and the base learner is XGBoost. For both the moderate and high noise settings, calibration is performed using the Null2 distribution obtained under the low noise scenario (To simulate real-world data characteristics, we perform calibration using data with a low level of noise). What's more, we want to evaluate the error of the selected heterogeneous region. Here, we define the threshold error, for each run, we extracted the decision threshold defining the heterogeneous region and computed the average of the squared threshold values. If a variable was not selected or the threshold exceeded 1, we assigned a value of 1.

Tables 6 and 7, together with Figure 11, demonstrate that noise level influences the performance of the M-learner in the All scenario. At moderate noise levels, the rejection rate reaches 1, whereas both low and high noise levels result in no rejections across 100 replications. Notably, performance exhibits only a slight decline under higher noise conditions. Overall, these findings indicate that although noise introduces variability, its effect on model performance remains limited.

Table 6: The table summarizes the correct covariates in profiles in scenario All for different noise levels with XGB learner. $X^{(1)}/X^{(2)}$:Final profile contain $X^{(1)}/X^{(2)}$, $X^{(1)}\&X^{(2)}$: final profile contain both $X^{(1)}$ and $X^{(2)}$, $X^{(1)}$ and $X^{(2)}$ are the two variables associated with treatment effect heterogeneity. Each value denotes the count of occurrences across 100 simulations.

| Covariates | $X^{(1)}$ | $X^{(2)}$ | $X^{(1)}\&X^{(2)}$ |
|---|---|---|---|
| Low | 100 | 100 | 100 |
| Moderate | 96 | 98 | 95 |
| High | 91 | 94 | 85 |

Table 7: The table summarizes the boundary of selected heterogeneous regions in profiles in scenario All for different noise levels with XGB learner. Values are the mean threshold errors over 100 replications, with standard deviations shown in parentheses. Threshold error: the decision threshold defining the heterogeneous region and computed the average of the squared threshold values. If a variable was not selected or the threshold exceeded 1, we assigned a value of 1.

| Covariates | $X^{(1)}$ | $X^{(2)}$ |
|---|---|---|
| Low | 0.056(0.194) | 0.059(0.162) |
| Moderate | 0.043(0.173) | 0.047(0.153) |
| High | 0.111(0.285) | 0.106(0.289) |

### A.4.3 Comparison of projection methods

In this section, we explain why we use t-SNE as projection method.

The core idea of our M-learner algorithm is that individuals with similar indirect treatment effects should be considered more similar. Based on this intuition, we construct a pairwise CAITE distance matrix that reflects heterogeneity between any two individuals. We then use t-SNE to project this matrix into a Euclidean space for clustering.

t-SNE (similarly UMAP) is particularly suitable in our method because it is designed to preserve local pairwise similarities during projection, which aligns with our goal of preserving treatment effect similarity. Unlike PCA (Maćkiewicz & Ratajczak, 1993), which performs linear projections and assumes Euclidean structure in the original space, our distance matrix is derived from treatment effect heterogeneity—not raw features—making non-linear methods like t-SNE or UMAP more

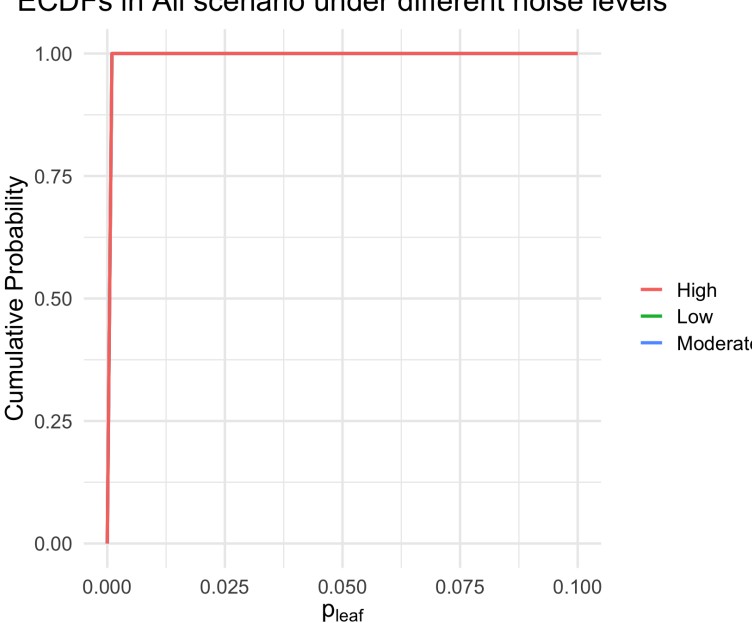

Figure 11: Empirical cumulative distribution functions (ECDF) of $p_{leaf}$ under different noises levels in All scenario.

appropriate. Notably, in our study, t-SNE directly operates on pairwise distances, which further justifies its application. In our study, t-SNE is employed to project the treatment effect distance matrix onto a low-dimensional Euclidean space while preserving local neighborhood structures. This approach facilitates the effective identification of potential subgroups, which are subsequently extracted through clustering.

Moreover, we also conducted sensitivity analyses using alternative projection methods such as UMAP(the goal in this step is not to reduce dimensionality per se, but rather to project the treatment distance matrix into a space that preserves their relative distances for downstream clustering. PCA focuses on preserving global variance rather than local or relational structure, it is not suitable for this purpose), and the results remained qualitatively consistent.

From Table 8 and Figure 12, these results support the t-SNE shows superior or comparable performance to UMAP across scenarios.

Table 8: The table summarizes the correct covariates in profiles in each scenario. $X^{(1)}/X^{(2)}$:Final profile contain $X^{(1)}/X^{(2)}$, $X^{(1)}\&X^{(2)}$: final profile contain both $X^{(1)}$ and $X^{(2)}$, $X^{(1)}$ and $X^{(2)}$ are the two variables associated with treatment effect heterogeneity. Each value denotes the count of occurrences across 100 simulations.

| | UMAP | | | t-SNE | | |
|---|---|---|---|---|---|---|
| Covariates | $X^{(1)}$ | $X^{(2)}$ | $X^{(1)}\&X^{(2)}$ | $X^{(1)}$ | $X^{(2)}$ | $X^{(1)}\&X^{(2)}$ |
| All | 97 | 98 | 96 | 100 | 100 | 100 |
| Part | 100 | 99 | 99 | 100 | 100 | 100 |
| Null1 | 6 | 2 | 2 | 8 | 4 | 2 |
| Null2 | 2 | 3 | 0 | 2 | 1 | 0 |
| Global | 1 | 3 | 1 | 3 | 5 | 1 |

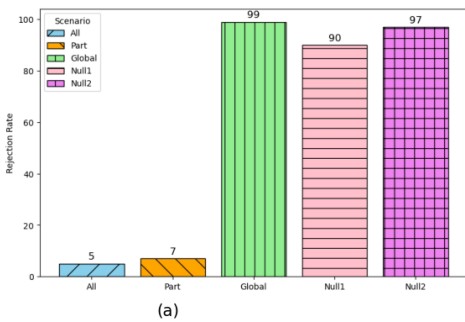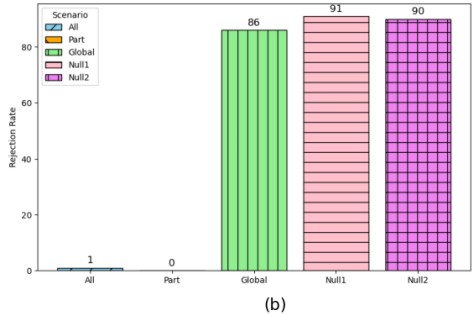

Figure 12: This table reports, after calibration, the number of rejections across 100 replications for each scenario, corresponding to the cases where no heterogeneous subgroups are identified. (a) UMAP, (b) t-SNE .

### A.4.4 DIMENSION OF PROJECTION

In this subsection, we use experimental results to illustrate why the projection matrix has a dimensionality of two.

Here, we evaluate the robustness of the dimension of projection. t-SNE only support reduce dimension to 2 or 3. Here we compare the results in All with different dimension of projections. For UMAP, it support reduce to higher dimension, we compare the results with $2, 5$ and $10$ in All scenario with XGboost learner.

As shown in the Table 9, the performance of t-SNE appears insensitive to the choice of projection dimension, while UMAP exhibits a clear dependence on it. In particular, UMAP achieves optimal performance when the projection dimension is set to 5, outperforming both the 2- and 10-dimensional settings. Furthermore, even the best t-SNE result is consistently outperformed by UMAP under its optimal configuration. Consequently, projecting the treatment distance matrix into two dimensions via t-SNE preserves the method's performance.

Table 9: The table summarizes the correct covariates in profiles in each scenario. $X^{(1)}/X^{(2)}$:Final profile contain $X^{(1)}/X^{(2)}$, $X^{(1)}\&X^{(2)}$: final profile contain both $X^{(1)}$ and $X^{(2)}$, $X^{(1)}$ and $X^{(2)}$ are the two variables associated with treatment effect heterogeneity. Each value denotes the count of occurrences across 100 simulations.

| Covariates | $X^{(1)}$ | $X^{(2)}$ | $X^{(1)}\&X^{(2)}$ |
|---|---|---|---|
| t-SNE(2) | 100 | 100 | 100 |
| t-SNE(3) | 100 | 100 | 100 |
| UMAP(2) | 97 | 98 | 96 |
| UMAP(5) | 99 | 100 | 99 |
| UMAP(10) | 94 | 94 | 89 |

### A.4.5 NUMBER OF MAXIMUM CLUSTERS

To investigate the impact of the maximum number of clusters on the results, we evaluate the effect of setting the maximum cluster number to 2, 5, and 10 in both the All scenarios. From Table 10 and Figure 13, we observe that the maximum number of clusters $k$ influences the results to some extent, but this effect becomes negligible once $k$ exceeds a certain threshold. Therefore, in practice, we recommend choosing an appropriate value of $k$ based on computational efficiency. To ensure optimal performance, we suggest selecting the largest feasible $k$ within resource constraints. In this paper, we recommend to select $k$ as $k = \left\lfloor \sqrt{d} \right\rfloor + 2$.

Table 10: The table summarizes the correct covariates in profiles in All cenario for different maximum clusters $K$. $X^{(1)}/X^{(2)}$:Final profile contain $X^{(1)}/X^{(2)}$, $X^{(1)}\&X^{(2)}$: final profile contain both $X^{(1)}$ and $X^{(2)}$, $X^{(1)}$ and $X^{(2)}$ are the two variables associated with treatment effect heterogeneity. Each value denotes the count of occurrences across 100 simulations.

| Covariates | $X^{(1)}$ | $X^{(2)}$ | $X^{(1)}\&X^{(2)}$ |
|---|---|---|---|
| $K = 2$ | 87 | 83 | 75 |
| $K = 5$ | 100 | 100 | 100 |
| $K = 10$ | 100 | 100 | 100 |

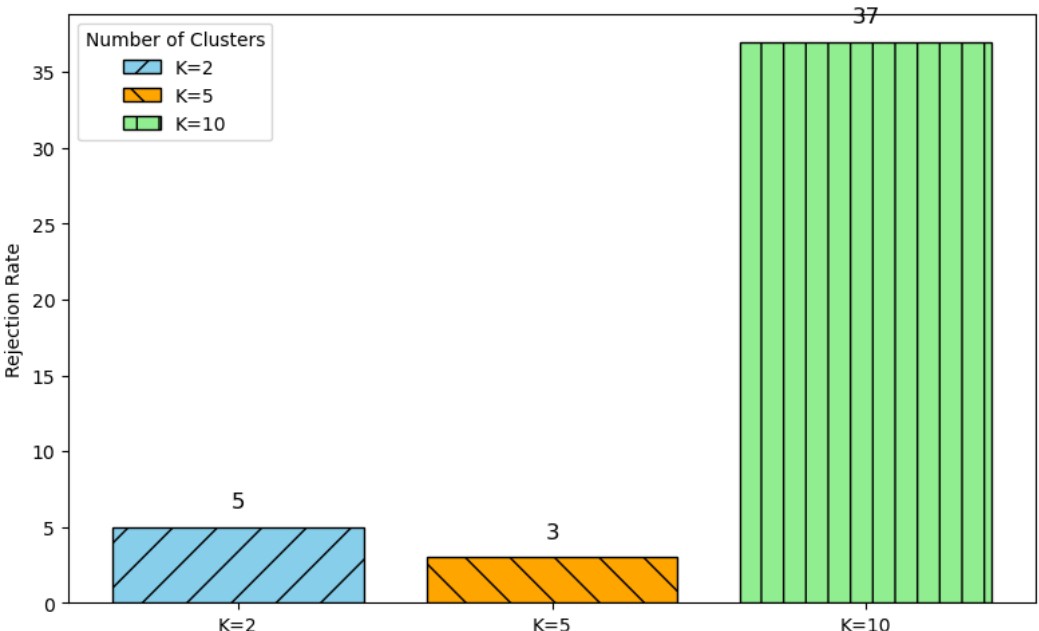

Figure 13: This table reports, after calibration, the number of rejections across 100 replications for different maximum clustering numbers $k$ for All scenario, corresponding to the cases where no heterogeneous subgroups are identified.

## A.5 VISUALIZATION

In this section, we visualize an intermediate step of the M-learner algorithm—specifically, the estimation of treatment effects—which provides insights into the underlying mechanisms contributing to its effectiveness.

When model with a mediator, we visualize the estimation of CAITE $\hat{\tau}^{ITE}(x) = \hat{g}_1^Y(x, \hat{g}_1^M(x)) - \hat{g}_1^Y(x, \hat{g}_0^M(x))$.

For each scenario , we estimate $\hat{g}_1^Y, \hat{g}_1^M(x)$ and $\hat{g}_0^M(x)$ using randomized clinical trial data. To facilitate visualization, we construct a synthetic grid of covariates where Cov1 and Cov2 vary from $-1.5$ to $1.5$ in increments of $0.05$, while all other covariates are drawn from a standard normal distribution. The estimated functions are then applied to this grid to compute the CAITE for each unit. We repeat this process across 100 simulated experiments and compute the average CAITE at each grid point. The resulting surface is interpolated using the R package "akima" and visualized to illustrate the spatial patterns in treatment heterogeneity. We visualize the results under a sample size of 1000 for each of the five scenarios—All, Part, Null1, Null2 and Global—using Random Forest and XGBoost as base learners. The visualizations are presented in Appendix Figure 14 - 18. Cov1 represents $X^{(1)}$, Cov2 represents $X^{(2)}$ in the following results.

According to the predefined settings of each scenario in , the heterogeneous region in the All and Part scenarios exhibit positive CAITE values. The Global scenario demonstrates uniformly positive

CAITE across the entire covariate space, whereas the Null1 and Null2 scenarios yield CAITE values consistently close to zero. Our visualizations effectively capture these patterns, providing a clear and accurate reflection of treatment effect heterogeneity across scenarios.

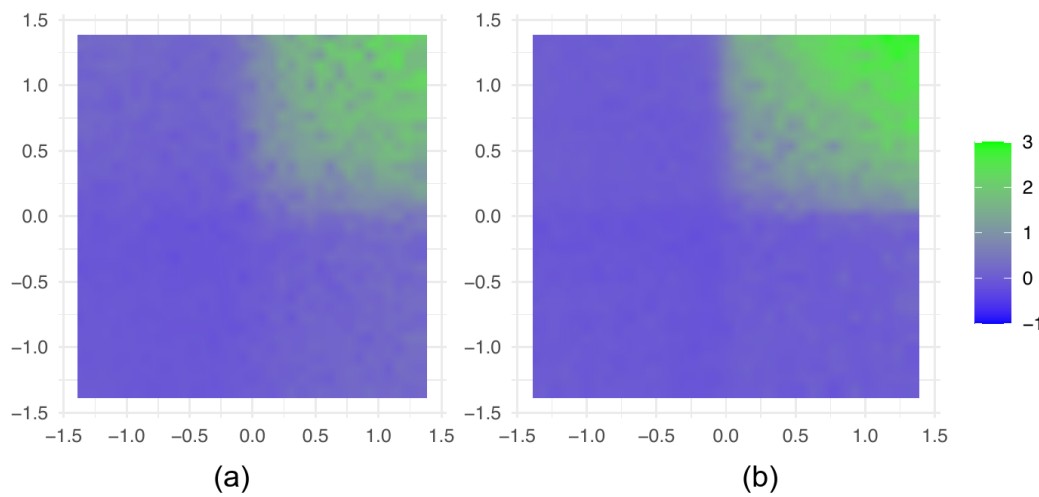

Figure 14: Visualization of CAITE in the All scenario:each panel plots Cov1 (x-axis,$-1.4$ to $1.4$) and Cov2 (y-axis, $-1.4$ to $1.4$), with color representing the estimated CAITE magnitude.(a) visualization of RF learner result, (b) visualization of XGB learner result.

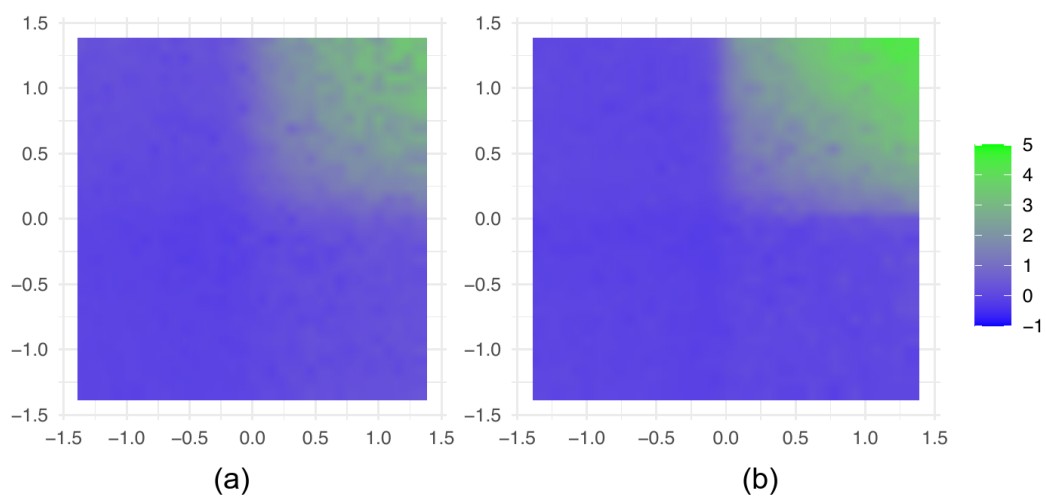

Figure 15: Visualization of CAITE in the Part scenario:each panel plots Cov1 (x-axis,$-1.4$ to $1.4$) and Cov2 (y-axis, $-1.4$ to $1.4$), with color representing the estimated CAITE magnitude.(a) visualization of RF learner result, (b) visualization of XGB learner result.

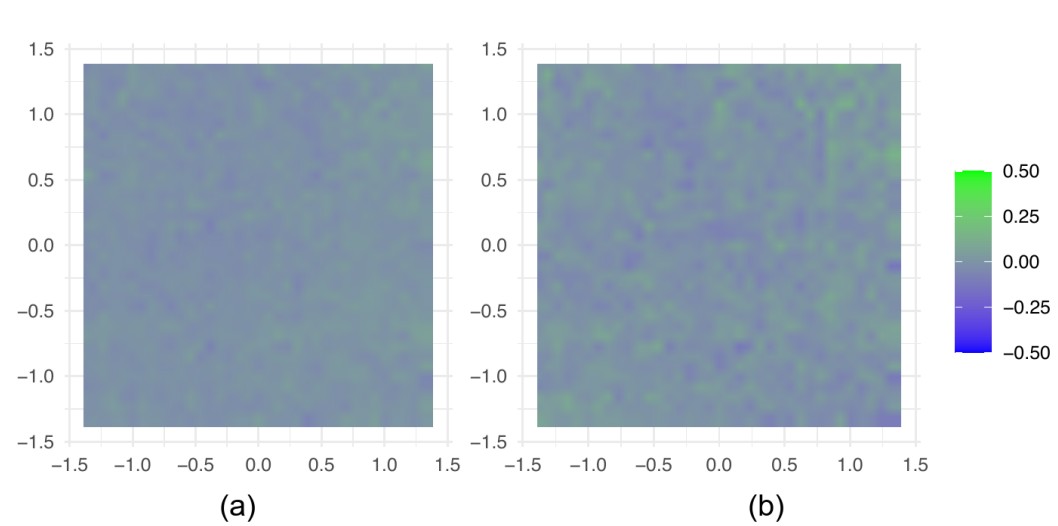

Figure 16: Visualization of CAITE in the Null1 scenario:each panel plots Cov1 (x-axis,−1.4 to 1.4) and Cov2 (y-axis, −1.4 to 1.4), with color representing the estimated CAITE magnitude.(a) visualization of RF learner result, (b) visualization of XGB learner result.

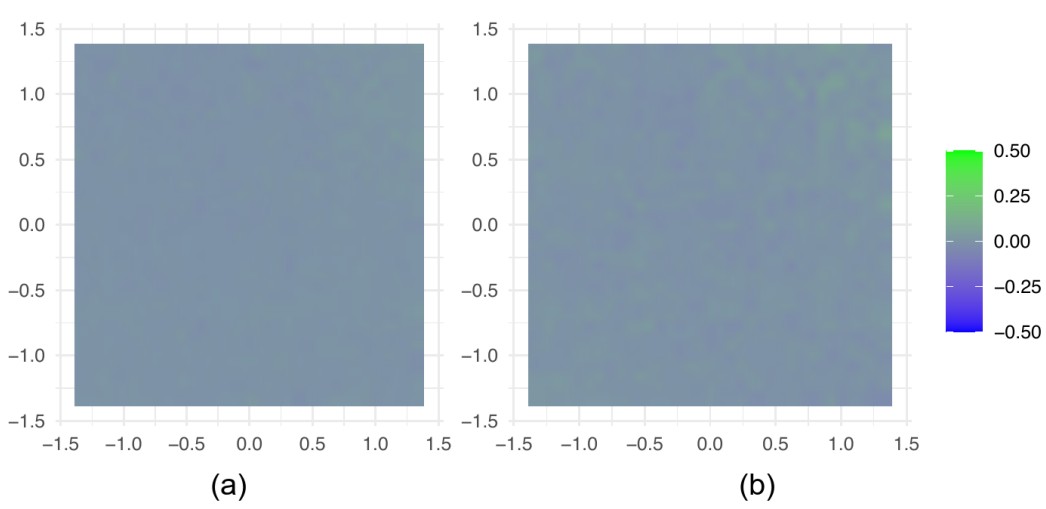

Figure 17: Visualization of CAITE in the Null2 scenario:each panel plots Cov1 (x-axis,−1.4 to 1.4) and Cov2 (y-axis, −1.4 to 1.4), with color representing the estimated CAITE magnitude.(a) visualization of RF learner result, (b) visualization of XGB learner result.

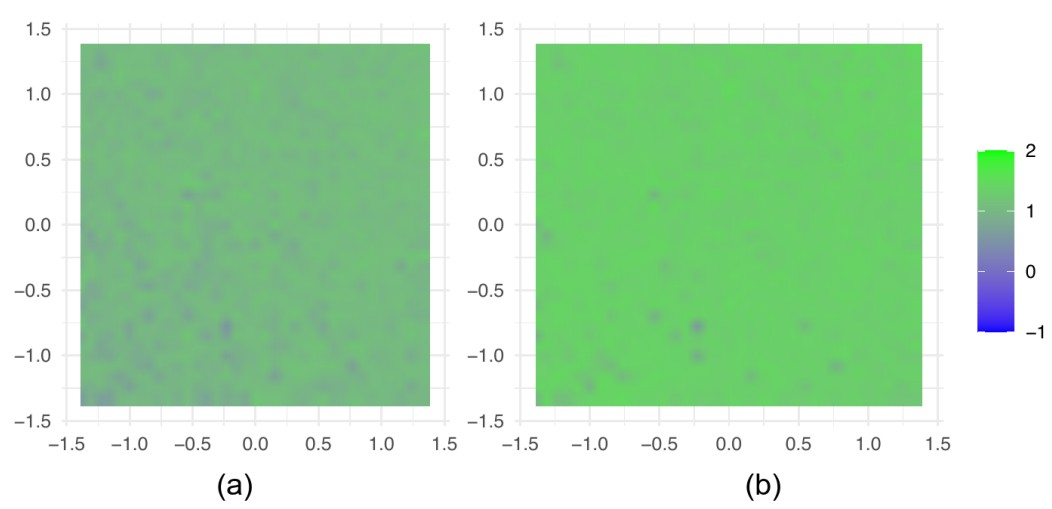

Figure 18: Visualization of CAITE in the Global scenario:each panel plots Cov1 (x-axis,−1.4 to 1.4) and Cov2 (y-axis, −1.4 to 1.4), with color representing the estimated CAITE magnitude.(a) visualization of RF learner result, (b) visualization of XGB learner result.

