# OpenReview forum: "M-learner: A Flexible And Powerful Framework To Study Heterogeneous Treatment Effect In Mediation Model"
_ICLR.cc/2026/Conference — ICLR 2026 Conference Withdrawn Submission_

### Official Review · Reviewer_drzE · 2025-10-26

**Soundness:** 2
**Presentation:** 2
**Contribution:** 2
**Rating:** 2
**Confidence:** 4

**Summary:**

This study proposes a method to estimate the natural indirect effect, focusing on a conditional estimation to then assess an analysis of the heterogeneity of this effect, depending on the covariates. The authors then propose a heuristic to derive interpretable subgroups of observations that share a similar indirect effect, to identify subgroups that most benefit the treatment.

**Strengths:**

The authors address an interesting question, with many practical implications, including the use of surrogate outcomes in clinical trials (in which case the surrogate outcome is the mediator). The authors also propose an end-to-end framework for analysis, which is always interesting to illustrate that the use case they propose is feasible.

**Weaknesses:**

The propose approach is not compared to any baseline approach. A lot of steps are very empirical, and they make sense in an applicative setting, but most steps will not have a theoretical grounding. In that case, an extensive empirical analysis, and comparison to alternative choices is needed, much stronger than the one proposed in this study.

**Questions:**

- a lot of existing mediation analysis estimators are proposed. In their implementation, most of them actually estimates conditional effects, that are then averaged over the individuals to provide average effects. The justification to develop a new estimation method is not clear, and a comparison to existing methods is absolutely necessary, even if those approaches are not explicitely framed to estimate the CAITE, they do in practice. It would be absolutely key to the strength of this study to add comparisons to those methods.
- Similarly, there are numerous works exploring the question of subgroups identification (or quantification of the effect heterogeneity) for the treatment effect not in the frame of mediation. Those would constitute a lot of baseline approaches to mention and compare the results to (one of them is Sverdrup, Erik, Maria Petukhova, and Stefan Wager. "Estimating treatment effect heterogeneity in Psychiatry: A review and tutorial with causal forests." International Journal of Methods in Psychiatric Research 34.2 (2025): e70015.)
- the authors claim that a possible application would be to have surrogate markers not to wait for the final outcome of the study to occur. Are there any guarantees that the sugroups for the indirect effect are the optimal subgroups for the total effect? is there an assumption regarding this aspect in your work?
- there are a number of minor formatting issues, with a lot of missing spaces around citations (l. 37, 52, 169, 170, 370...), l.186 there is an opened quote sign with the rest of the test on the line after
- there seems to be a missing condition in equation 1, about the intersection of U^{ITE}_i and U^{ITE}_j
- in Figure 5, the caption refers to the variable econ_hard, but I d'ont see it in the tree. It is really hard to read the figure to actually interprete the type of individuals in the two subtype, so that is a shame for an interpretable result. Improvement of this figure would be welcome.

The work of providing an extensive comparison to the existing approaches for the first two points is really important, and is probably hard to fit within the rebuttal phase, hence my rating.

---

### Official Review · Reviewer_XjGn · 2025-10-28

**Soundness:** 3
**Presentation:** 2
**Contribution:** 3
**Rating:** 4
**Confidence:** 3

**Summary:**

The paper proposes M-learner, a machine learning–based framework for estimating heterogeneous indirect treatment effects within mediation analysis. The method estimates individual Conditional Average Indirect Treatment Effects (CAITE), constructs pairwise distances, applies t-SNE and K-means clustering to identify subgroups, and uses a likelihood ratio–based calibration (pleaf) to control Type I error. Experiments on synthetic data under multiple heterogeneity scenarios and a real-world JOBS II dataset demonstrate that M-learner can effectively detect valid subgroups and maintain statistical control.

**Strengths:**

1. The paper addresses an important and underexplored problem—estimating heterogeneous indirect treatment effects in mediation analysis.

2. The proposed M-learner framework is conceptually clear and integrates machine learning flexibility with statistical inference for subgroup discovery.

3. The method provides an explicit mechanism (via pleaf) to control Type I error, which is promising.

4. The framework is model-agnostic, allowing the use of different base learners such as Random Forests and XGBoost, enhancing its general applicability.

**Weaknesses:**

1. The method relies on the random treatment assignment assumption and is therefore limited to RCT data, reducing applicability to observational studies.

2. The consistency theorem is presented without a rigorous proof or asymptotic error analysis, making the theoretical foundation somewhat weak.

3. The t-SNE + K-means clustering step introduces randomness and hyperparameter sensitivity, which may affect subgroup stability and reproducibility.

4. The calibration procedure based on p_leaf lacks detailed justification and description, however this is quite important for the readability of this paper.

5. The comparison with the BART-based method (Ting & Linero, 2025) in the appendix is limited and does not provide comprehensive quantitative evaluation.

**Questions:**

The following is my questions and concerns:

1. The introduction primarily reviews the HTE literature, with insufficient discussion on the definition and challenges (such as identification and estimation biases) of heterogeneous mediation effects.

2. It's unclear why the existing methods can not be adapted to mediation settings by modeling the mediator as part of covariates or outcome mechanisms, and then by stratification to analyse? For example, taking the mediator as the outcome variable.

3. Selecting valid mediator variable is quite important for this work. It's unclear in the intro or method section if the mediator is pre-defined or known? If no, how to identify a valid mediator? If valid mediator is known, existing HTEs methods can be directly adapted to mediator setting with replacing the definition of outcome variable.

4. The decomposition TTE=ITE+DTE mplicitly assumes additive effects; how does the method handle potential interaction between direct and indirect pathways?

5. The main limitation of the proposed M-Learner framework is that it relies on the assumption of random treatment assignment, making it applicable only to randomized controlled trial (RCT) data. In most real-world observational studies, treatment assignment is confounded with covariates or unobserved factors, violating this assumption and potentially biasing the estimation of CAITE and subgroup identification. Since the method lacks confounding adjustment mechanisms (e.g., propensity weighting or doubly robust estimation), its identifiability and validity cannot be guaranteed beyond RCT settings.

6. Furthermore, Assumption 3 (mediator ignorability) is particularly strong and generally unverifiable even under randomization, since unmeasured mediator–outcome confounders often exist even in RCT.

7. Eq (2) and (3) are not clearly stated. What is $\eta(x)$ and $k(x)$? what do they represent in a mediation model?

8. In theorem 2.1, the transition from the integral expression to $g_1^Y(x,g_1^M)-g_1^Y(x,g_0^M)$ is non-trivial and seems to assume linear or mean-value properties that may not hold.

9. Since CAITE is scalar, why not cluster directly on τ^\hat (e.g., 1D k-means or optimal thresholding) instead of t-SNE+K-means?

10. the selection of final subtype classification is not clear. what does the each leaf of decision tree actually mean? What does p_leaf represent? Why construct a decision tree to model the clustering results? Why using the minimal p_leaf as the final results? All theses issues are not clearly stated, making the methodology hard to read. The paper should clearly state the process of how to model M by decision tree (eq 4 and 5) and how to use the test to select the final subtype.

11. The setting of the experiment is really unclear and confusing. For example in table 1, why report the X(1) and X(2)? They are the ground truth?

12. why the comparison with existing heterogeneous mediation methods (e.g., BART-based Ting & Linero, 2025 and more) is not reported in the main body?Furthermore, The comparison with the BART-based model presented in the appendix is not sufficiently comprehensive. The authors should provide more empirical evaluation adopting the same setting and experiment design as the ones in main body.

---

### Official Review · Reviewer_4JSb · 2025-10-31

**Soundness:** 2
**Presentation:** 2
**Contribution:** 1
**Rating:** 2
**Confidence:** 5

**Summary:**

The paper proposes a new framework, called the M-learner, designed to estimate heterogeneous indirect and total treatment effects and to identify subgroups with similar mediation patterns. The proposed method first estimates conditional average indirect and total treatment effects for each individual, and then, it constructs a pairwise distance matrix capturing heterogeneity among these effects. Next, it applies t-SNE to project the distance matrix into a low-dimensional space and performs K-means clustering to detect subgroup structures. Finally, it refines the clusters through a threshold-based calibration to determine the optimal grouping. In this way, the method offers a way to uncover complex interdependencies among covariates, mediators, and outcomes within mediation analysis while purportedly maintaining control over Type I error. Although the topic, heterogeneous mediation effects, is of potential interest, this submission does not offer sufficient originality, rigor, or theoretical contribution to merit publication at ICLR.

**Strengths:**

- The proposed method offers a practical pipeline for mediation heterogeneity. Clear, modular steps make it easy to implement and plug into existing learners, enabling subgroup discovery focused on indirect (mediated) effects.
-  Simulations and the JOBS II case illustrate how the method can flag groups where the mediator is (or isn’t) operative, suggesting use for interim monitoring or targeted interventions in trials.

**Weaknesses:**

1. My main hesitation is the fact that The proposed method is primarily a pipeline of existing techniques (e.g., standard causal mediation decomposition, off-the-shelf machine learning estimators, t-SNE projection, and K-means clustering), with limited theoretical depth and weak empirical validation. The 'M-learner' seems just a combination of known components: basic causal mediation model, CAITE estimation, t-SNE projection, and K-means clustering. Given that there is no new causal identification result, the algorithm’s novelty is rather superficial.

2.  There is no formal analysis of consistency, convergence, or error bounds for the discovered heterogeneous subgroups, despite prior top-tier work on related problems providing strong guarantees.

3. The work does not engage with modern causal learning theory, e.g., semiparametric efficiency, double robustness, orthogonalization, or debiasing. Moreover, references to recent methods (R-learner, DR-learner, T-learner, etc.) are mentioned yet not properly contrasted. There is no clear argument why this framework improves over them or why mediation heterogeneity cannot be addressed by existing meta learner extensions.

4. Methodological choices seems arbitrary. I believe the use of t-SNE for identifying causal subgroups is methodologically inappropriate; it distorts global distances and cannot guarantee meaningful subgroup structure. Also, 'Calibration via p-leaf' is ill-defined and lacks statistical justification. At the very least, the authors provide sound argument over these choices.

5. The paper is verbose and repetitive, mixing conceptual explanation with implementation details.

**Questions:**

Could you provide a convincing, point-by-point response to the critiques above?

---

### Official Review · Reviewer_Zc3W · 2025-11-03

**Soundness:** 1
**Presentation:** 2
**Contribution:** 1
**Rating:** 2
**Confidence:** 4

**Summary:**

In this article, an estimation method for indirect under a mediation framework. The procedure comprises four key steps.
In addition to estimating mediated heterogeneity, an approach for identifying subgroups is proposed.
Simulation and empirical studies are provided.

**Strengths:**

The paper is well-motivated.

**Weaknesses:**

1. The paper requires substantial revision to provide more rigorous validation of its underlying assumptions. Assumptions 3 and 4, in particular, lack sufficient substantiation. In practice, the precise identification of mediators remains elusive, and the method's reliance on numerous unobserved elements renders its applicability questionable. Furthermore, the data-driven approach to subgroup identification appears fundamentally weak, as the paper conspicuously neglects recent advances in semi-parametric efficient methodologies.

2. The paper needs to be clear about how the type I and type II errors are controlled. I cannot find the evidence other than numerical studies, which can be easily manipulated, and theoretical justification are needed if the claim remains in the paper.

**Questions:**

- Please see **Weakness**.

---

### Note · Authors · 2025-11-30

**Comment:**

The method proposed in this manuscript is designed to address the heterogeneity of surrogate performance in clinical trials. In practice, surrogate biomarkers (mediators) often exhibit substantial heterogeneity across patient populations, which can critically affect trial conclusions. Notably, 31 drugs that were granted accelerated approval based on surrogate biomarkers were subsequently withdrawn by the FDA, underscoring the urgency of this challenge for the pharmaceutical industry. Our proposed approach offers a potential solution to this problem. We appreciate the reviewers’ insightful comments and will revise the manuscript accordingly in the future.

**Withdrawal Confirmation:**

I have read and agree with the venue's withdrawal policy on behalf of myself and my co-authors.